# Evidence-Based Physiotherapy Practice in Paediatric Subdiscipline: A Cross-Sectional Study in Saudi Arabia

**DOI:** 10.3390/healthcare10112302

**Published:** 2022-11-17

**Authors:** Mshari Alghadier, Ragab K. Elnaggar, Muneera I. Alasraj, Najwa Khan, Aseil Almeiman, Reem Albesher

**Affiliations:** 1Department of Physical Therapy and Health Rehabilitation, College of Applied Medical Sciences, Prince Sattam Bin Abdulaziz University, Alkharj 11942, Saudi Arabia; 2Department of Physical Therapy for Pediatrics, Faculty of Physical Therapy, Cairo University, Giza 12613, Egypt; 3Saudi Pediatric Physical Therapy Group, Saudi Physical Therapy Association, Riyadh 12372, Saudi Arabia; 4Department of Physical Therapy, Rehabilitation Hospital, King Fahad Medical City, Riyadh 12231, Saudi Arabia; 5Department of Rehabilitation Science, College of Health and Rehabilitation Sciences, Princess Nourah bint Abdulrahman University, Riyadh 11671, Saudi Arabia

**Keywords:** evidence-based practice, physiotherapy, decision making, healthcare, Saudi Arabia, rehabilitation, paediatric

## Abstract

This cross-sectional study explored the behaviour, knowledge, skills and resources, opinion, and perceived barriers of paediatric physiotherapists practising in Saudi Arabia regarding evidence-based practice (EBP). Sixty-eight paediatric physiotherapists from Saudi Arabia participated. Data were collected by electronic questionnaire and the Likert scale was used to score knowledge, skills and resources, opinion, and barriers to EBP implementation. Approximately 78% of the participants were motivated to use EBP in their daily practise and 82.3% have reported the use of best scientific evidence in their clinical practise. Participants with higher database usage over the last 6 months showed significant association with EBP knowledge scores (t = 2.46, *p* = 0.01), skills and resources scores (t = 3.81, *p* < 0.001), and opinion scores (t = 2.43, *p* = 0.01). Furthermore, a higher level of education in participants was significantly associated with EBP knowledge scores (t = 2.41, *p* = 0.01). Most paediatric physiotherapists believed that EBP is essential in their clinical practise as it improves patient care and quality of health services. Difficulty in obtaining full-text papers and lack of time were identified as major barriers to implementing EBP followed by the lack of management support, motivation in research, and EBP training.

## 1. Introduction

Translating research findings into clinical practise is not straightforward. Clinicians sometimes fail to base their clinical practise on scientific research, and academic research may be irrelevant in clinical practise [1]. Evidence-based practice (EBP) has been introduced to fill the gap between clinical practise and scientific research. EBP includes integrating clinical expertise, patient’s values and preferences, and best available research evidence to enhance clinical decision-making processes and achieve high-quality patient care [2,3].

Physiotherapy is one of the professions that has limited scientific support, and the focus on encouraging EBP to move away from interventions has increased. American Physical Therapy Association (APTA) vision 2020 statement has suggested that EBP is an important goal for this field [4]. Several studies have investigated the effectiveness of EBP learning theories for undergraduate students; however, there is no conclusive evidence for the most effective theory for developing and supporting EBP capability [5,6]. Although cumulative evidence indicates that physiotherapists have a positive attitude regarding EBP and its implementation, further education and training are required in this field [3,7,8]. A recent systematic review (2021) identified barriers to implementing EBP in physiotherapy. In addition to the language barrier, commonly reported barriers include lack of time to discover and appraise evidence; skills to search, identify, and apply the best evidence; and access to the evidence [8]. Further barriers include a low perception of EBP, lack of interest or support from employers, statistical understanding [7], and educational barriers [8]. The literature clearly shows that clinical experience is currently involved in the EBP approach [2,9].

The implementation of EBP is encouraged to improve costs in the presence of abundant information and customers’ awareness of treatment decisions. Additionally, it provides an opportunity to make treatment personalised, more effective, streamlined, and dynamic; and to optimise clinical decision making. EBP enhances best practise, improves service quality [10], minimises delay, and helps to balance advantages and disadvantages in clinical practice [11]. In recent years, there has been a rapid expansion of evidence to support paediatric physiotherapists in choosing safer and efficient interventions [12,13,14,15,16]. While there is strong evidence supporting paediatric clinical practice including early identification of children at high risk for cerebral palsy, there is little published literature on the implementation of these guidelines into clinical practice [17].

The field of physiotherapy is relatively new in Saudi Arabia, with the first bachelor’s program established in 1985 at King Saud University [18], which is later than other countries [19,20,21]. However, there has been a massive expansion in physiotherapy over the last two decades, with more than 16 universities teaching physiotherapy programmes across the country [19]. According to the Saudi Commission for Health Specialities (SCFHS), there are 6028 registered physiotherapists in Saudi Arabia in 2018 [22]. Physiotherapy subspecialties in Saudi Arabia are musculoskeletal/orthopaedic, neurology, cardiopulmonary and vascular, sports, women’s health, and paediatrics [23]. There is no accurate reference number of paediatric physiotherapists in Saudi Arabia.

A number of previous studies have reported positive attitudes of physiotherapists in Saudi Arabia regarding EBP and its implementation [24,25,26]; however, no studies exploring EBP and Saudi paediatric physiotherapy have been completed to date. In 2017, a survey of 376 physiotherapists found most respondents did not receive formal EBP training (70%) in universities or any other authorised training centres [25]. The most frequently reported barriers to the implementation of EBP in Saudi Arabia include insufficient teaching in previous education and lack of research knowledge and skills [25]. The results of Hasani, et al. (2020) showed that among physiotherapists practising in Saudi Arabia, EBP had not been extensively implemented; however, positive attitudes regarding its implementation have been expressed [24].

The available evidence highlights a prominent gap in understanding and applying the EBP concept among paediatric physiotherapists practising in Saudi Arabia. However, to the best of our knowledge, EBP among paediatric physiotherapists have yet to be investigated in Saudi Arabia. Therefore, this study aimed to explore the attitudes, knowledge, and implications of EBP among paediatric physiotherapists practising in Saudi Arabia. This aimed to help bridge gaps and build future recommendations to improve the practise and implication of EBP among paediatric physiotherapists in Saudi Arabia.

## 2. Materials and Methods

This study used a cross-sectional descriptive study design. The Research Ethics Committee of Prince Sattam bin Abdulaziz University, Alkharj, Saudi Arabia approved this study (No.: RHPT/022/007). Participants read and electronically accepted the participant information and consent forms before answering the questionnaire.

This study recruited a convenience sample of physiotherapists in the paediatric subdiscipline working in Saudi Arabia. The inclusion criteria were paediatric physiotherapists working in a clinical or academic setting in Saudi Arabia. The exclusion criteria included paediatric physiotherapists working abroad, interns and students.

The questionnaire was previously implemented with physiotherapists from Brazil and UAE [27,28]. The questionnaire was tested previously in pilot study with Portuguese language to ensure understanding and quality [29]. It was categorised into the following domains: (1) consent form (1 item); (2) demographic details, educational, and professional experience (13 items); and (3) characteristics relating to EBP behaviour (4 items)—knowledge (9 items), skills and resources (9 items), opinion (5 items), and barriers (17 items). The questionnaire was developed using response options on a 5-point Likert scale between one and five (1 = strongly disagree, 2 = partially disagree, 3 = neutral, 4 = partially agree, and 5 = strongly agree). However, for the negative statement items in the questionnaire (knowledge statement 2, and opinion statement 2, among others), the value rating was reversed (1 = strongly agree, 2 = partially agree, 3 = neutral, 4 = partially disagree, 5 = strongly disagree).

The online platform Google Forms was used to complete the questionnaire and collect the data. It is a free service tool for creating online forms. The questionnaire spread started on 7 April 2022 and concluded on the 8 July 2022. In addition, the questionnaire was distributed through social media platforms (Twitter and WhatsApp). The Saudi Physical Therapy Association (SPTA) Paediatric subgroup emailed an electronic questionnaire to all members and advertised the study on its website. The advertisement provided a link to the questionnaire page. All information collected was anonymous to guarantee maximum confidentiality, and only the authors had access to the data.

The data were automatically generated from the online questionnaire to an Excel spreadsheet, and statistical analyses were conducted using R version 4.0.3 (2020-10-10). Data were analysed using descriptive statistics and reported as percentages, values, and frequencies. The mean and standard deviation (SD) were calculated for each item by transforming the 5-point Likert scale into numerical values ranging from 1 to 5 and then averaging the responses across participants to analyse the self-reported characteristics related to EBP. Subsequently, scores were generated for each scale (knowledge, skills and resources, opinion, and barrier scores) by summing up the points obtained for the corresponding questions related to the specific scale. The total score was calculated by summing up the four scales together. A higher total score indicated a more positive behaviour towards EBP. Finally, a simple correlation analysis was performed to determine whether there was any relationship between the generated scores and demographic variables. Pearson’s correlation coefficients were calculated, and the strength of the correlation was interpreted as very weak (0.0–0.2), weak (0.2–0.4), moderate (0.4–0.6), strong (0.6–0.8), and very strong (0.8–1.0). A multiple regression analysis was subsequently conducted, with different generated scores and demographic characteristics as dependent and independent variables, respectively. Statistical significance was set at *p* < 0.05.

## 3. Results

Overall, 68 paediatric physiotherapists participated in the study; most of them were female (75%), aged 21–30 years (50%), and with approximately 5 years of experience (56%). Almost all participants were Saudi nationals (85.2%), with the majority from the Riyadh region (67.6%). Approximately 62% of the participants only had an undergraduate degree and worked in a government setting –in both inpatient and outpatient facilities (44.1%). Most of them specialised in a neurodevelopmental subspecialty (72%) and had previously conducted or participated in research (64.7%) (Table 1).

The findings of paediatric physiotherapists’ behaviour towards the use of databases showed that the majority used PubMed (94.1%), Google Scholar (83.8%), PEDro (73.5%), and MEDLINE (54.4%). Approximately 35% and 28% of participants used online databases one to three times a month and one to three times a week, respectively. Respondents stated that they generally had a good understanding of the term EBP (77.9%), understood the core element of EBP (75%), and had sufficient knowledge to use EBP (58.8%). Most of them had previous experience with EBP during their undergraduate and postgraduate studies (82.5%), and approximately half assumed that the information provided during undergraduate courses was sufficient (48.6%). Approximately 60% of the respondents stated that they understood how to apply research findings to clinical practice and different study designs. Most respondents (80.8%) stated that they understood the statistical data, and 89.7% were interested in learning more about EBP (Table 2).

More than 80% of the respondents could conduct online database searches, with 54.4% being able to critically assess scientific data. Approximately 78% of the participants were motivated to use EBP in their daily practice, 82.3% used the best scientific evidence in their clinical practise, and 60.3% lacked discussion about EBP in their workplace. Most respondents had computer and internet access at the workplace (57.4%) and frequently accessed online databases (54.4%). Approximately 67.7% of the respondents asked their patients about their preferences in their clinical decision-making, and 75% of them informed their patients about treatment options (Table 3).

Approximately 80% of respondents think EBP is essential in their clinical practise; it improves patient care in physiotherapy (95.6%), and 54.4% think most of their decision-making incorporates EBP. In addition, most of the respondents (87%) believed that using scientific evidence would improve the quality of health services. In comparison, 35.3% assumed that expert opinion was the most crucial factor in their clinical decision-making (Table 4).

The most significant barrier to using EBP, as identified by 64% of respondents, is difficulty in obtaining full-text papers. Approximately 60% of the respondents rated the lack of time as the second major reason for not using EBP. Finally, lack of management support (56%), motivation in research (54.4%), and EBP training (51.4%) were rated as important barriers to EBP implementation (Table 5).

Scores were generated for each scale and are presented in Table 6.

The Pearson’s correlation analysis between the generated scores is presented in Table 7. When examining the correlation between sociodemographic characteristics and generated scores, age was significantly correlated with skill and resources (r = 0.36) and opinion (r = 0.35). Years of experience were significantly correlated with opinion (r = 0.35), and education level was significantly correlated with knowledge (r = 0.41), skills and resources (r = 0.37), and opinions (r = 0.27). Database usage frequency during the last 6 months was significantly correlated with knowledge score (r = 0.38), skills and resources (r = 0.50), and opinion (r = 0.34).

Multiple regression analysis with generated scores as an outcome variable revealed that the level of education and database usage over the last 6 months was significantly associated with EBP knowledge scores. Skills and resources, as outcome variables, demonstrated a significant association with database usage over the last 6 months. The opinion score was significantly associated with years of experience and database usage over the last 6 months. However, the barrier-generated score was not significantly associated with any variable (Table 8).

## 4. Discussion

This study explored the behaviour, knowledge, skills and resources, opinion, and perceived barriers of paediatric physiotherapists practising in Saudi Arabia regarding evidence-based practice. This is the first study to investigate physiotherapy paediatric subspecialty in Saudi Arabia and it adds to the limited body of knowledge available in physiotherapy subspecialties. The study’s results indicated that the knowledge and opinion score had the highest mean score, followed by skills and resources, and, finally, the barrier score. Therefore, this implies that paediatric physiotherapists have positive knowledge and opinions regarding EBP and perceived barriers that significantly influence their EBP application. In addition, our results found a significant association between some demographics (such as educational level and years of experience) and knowledge, skills, resources, and opinion scores.

### 4.1. Self-Reported Behaviour and Knowledge towards EBP

Our findings indicated that paediatric physiotherapists practising in Saudi Arabia have positive behaviour towards EBP and database usage, with PubMed being the most commonly used database, where more than 94% of participants used it; however, the Cochrane library was the least used (38.2% of participants). Approximately 35% of participants used the online databases one to three times a month, which is similar to previously published data from Brazil (32.7% of the dermatology subspecialists) [28], and Sweden (23% of primary health care physiotherapists) [30], and higher than in the UAE (15.2% of cardiopulmonary physiotherapists) [31].

Most of our sample reported that they understood the meaning of EBP and its core elements. However, approximately 52% believed that the information provided in undergraduate courses about EBP was insufficient. This is in contrast to a previously published study by Jette (2003) in the USA (82%) and similar to Canadian physiotherapists in neurology subspecialty (44.5%) and the Philippines (50%) [32,33]. Therefore, we believe this finding might highlight the need to integrate EBP into undergraduate courses and further invest in curriculum development and improvement in physiotherapy programmes in the country. Furthermore, almost 90% of the sample were interested in learning more about EBP which emphasises the necessity for training opportunities within educational institutions. As shown previously, EBP training is associated with higher implementation rates of research evidence in clinical practise [34], in addition to continuous professional education programmes related to EBP, which can be an effective strategy to support its adoption by clinicians [4].

### 4.2. Self-Reported Skills, Resources, and Opinion towards EBP

Although more than 80% of respondents reported that they use the best scientific evidence in their clinical practise, only 53% of them have reported that they can critically appraise scientific papers, which was higher than previously reported data from the UAE with 30% [31], and Australia with 26% [35]. Critical appraisal is one of the steps in EBP; therefore, the limited percentage of paediatric physiotherapists with critical appraisal skills and application indicates a lack of EBP implementation in clinical practise. However, high self-reporting of EBP skills and resources does not always reflect greater application [35]. Availability of computer resources and internet access in the workplace has been identified as a main enabler of EBP adoption [36]. Nevertheless, 44% of our participants did not have these resources, which might limit their EBP applicability in clinical practise. Allocating financial funds towards these infrastructure resources would play an integral role in enhancing EBP applicability, which will reflect on a better healthcare provision.

Almost 80% of the current study sample reported the importance of EBP in their clinical practise, and 90% believed it improved patient care in paediatric physiotherapy. Despite that, approximately 45% of them did not base their treatment decision-making on scientific research. We believe this, on one hand, reflects the notion that paediatric physiotherapists recognise EBP importance towards high-quality healthcare, while on the other hand, finds it challenging to incorporate it in their clinical practise. This might be due to misunderstanding or underestimation of their technical expertise related to EBP (e.g., critical appraisal, results interpretation, and statistical understanding). Expert opinion is recognised as the lowest level of evidence; however, approximately 35% of participants believed it is the most crucial factor in clinical decision-making which is lower than other results reported in the literature [27,28,37].

### 4.3. Self-Reported Barriers towards EBP

Workload pressure, obtaining full text, management support, and lack of time were the top reported barriers in this study which is consistent with previously published work [25,30,33,35,37,38,39]. However, workload pressure and lack of time are unlikely to ease in the future especially in paediatric physiotherapy subspecialties due to a limited number of specialists in Saudi Arabia. Therefore, decision makers should draw their attention to modifiable barriers such as obtaining full text, management support, interpreting results, and understanding statistics. Different strategies have been previously proposed to mitigate modifiable barriers faced by physiotherapists, such as educational workshops, discussion about EBP in the workplace, accessible and condensed summaries of evidence, and reimbursement and incentives for time to complete EBP activities [4].

### 4.4. Strengths and Limitations

To the best of our knowledge, this descriptive study reports, for the first time, data on the attitudes and behaviours regarding EBP and areas of strength and weakness in the current knowledge, skills, and barriers of paediatric physiotherapists in Saudi Arabia. Additionally, we had a satisfactory response rate and we used different resources to improve data collection by sending questionnaires through social media platforms and electronic advertising on the association website (with direct access to the questionnaire). This study indicated a positive attitude towards using EBP, with most participants responding agree or strongly agree (80%) to the importance of research in their clinical practise.

This study had a few limitations. First, the small sample size compared to the total number of physiotherapists practising in Saudi Arabia, which might have affected the current results. This limited participation might have been due to the short period of time (only three months to collect data) and limited number of physiotherapists registered with the professional organisation (SPTA). Second, the majority of the study sample were from Riyadh city (67.6%) compared to other cities of Saudi Arabia, which might impact the generalisability of the study findings. Third, the study was based on a self-reported questionnaire, which might have resulted in some subjective bias. The questionnaire was not assessed in a pilot study by the authors to measure its psychometric properties such as consistency and readability. However, it was previously developed in pilot study on Portuguese physiotherapists [29], and previously used by other researchers [27,28]. Finally, it is possible that therapists who are more confident in implementing EBP, are more likely to return surveys as approximately half of the study sample had research experience and a third had teaching experience.

## 5. Conclusions

This study explored the behaviour, knowledge, skills and resources, opinion, and perceived barriers of paediatric physiotherapists practising in Saudi Arabia regarding evidence-based practice. To our knowledge, this is the first study to explore EBP among paediatric physiotherapists in Saudi Arabia. The majority of paediatric physiotherapists believe EBP is essential in their clinical practise as it improves patient care. In addition, most of the respondents believed that using scientific evidence would improve the quality of health services. Difficulty in obtaining full-text papers and lack of time were identified as major barriers for implementing EBP followed by the lack of management support, motivation in research, and EBP training. The findings of this study have the potential to guide hospital management and academic staff in supporting the implementation of EBP in paediatric clinical practise and overcome potential barriers.

## Figures and Tables

**Table 1 healthcare-10-02302-t001:** Demographic characteristics of respondents (n = 68).

Characteristics	n (%)
**Gender**	
Female	51 (75)
Male	17 (25)
**Age**	
21–30	34 (50)
31–40	25 (36.7)
41–50	8 (11.8)
Above 50	1 (1.5)
**Nationality**	
Saudi	58 (85.2)
Non-Saudi	10 (14.8)
**Region**	
Riyadh	46 (67.6)
Makkah	7 (10.3)
Qassim	6 (8.8)
Jazan	4 (5.8)
Baha	2 (3)
Eastern	2 (3)
Aseer	1 (1.5)
**Years of experience**	
<2 years	19 (28)
2–5 years	19 (28)
6–10 years	15 (22)
11–15 years	8 (11.7)
>15 years	7 (10.3)
**Education level**	
Undergraduate degree	42 (61.8)
Postgraduate degree (master’s)	20 (29.4)
Postgraduate degree (PhD)	6 (8.8)
**Work setting**	
Academic	11 (16.2)
Government setting (inpatient and outpatient facility)	30 (44.1)
Private setting (inpatient and outpatient facility)	21 (30.9)
Homecare	6 (8.8)
**Subspecialty**	
Neurodevelopmental	49 (72)
Musculoskeletal	11 (16.1)
Neonatal	7 (10.5)
Haemophilia	1 (1.4)
**Teaching experience**	
Yes	36 (52.9)
**Research experience**	
Yes	44 (64.7)
**Professional membership**	
Yes, membership is important	31 (45.5)
No, not a member	37 (54.5)

**Table 2 healthcare-10-02302-t002:** Self-reported characteristics related to EBP—Knowledge.

Questions about Knowledge	Categories	N	%	M ± SD
I understand the meaning of the term EBP	Strongly disagree	4	5.9	4.26 ± 1.17
	Partially disagree	2	2.9	
	Neutral	9	13.3	
	Partially agree	10	14.7	
	**Strongly agree**	**43**	**63.2**	
I do not understand the core elements of EBP	**Strongly disagree**	**37**	**54.4**	1.77 ± 0.97
	Partially disagree	0	0	
	Neutral	14	20.6	
	Partially agree	4	5.8	
	Strongly agree	13	19.2	
I believe I have sufficient knowledge to use EBP	Strongly disagree	5	7.4	3.60 ± 1.22
	Partially disagree	8	11.8	
	Neutral	15	22.1	
	**Partially agree**	**21**	**30.9**	
	Strongly agree	19	27.9	
I had no experience with EBP during my undergraduate or postgraduate course(s)	**Strongly disagree**	**33**	**48.5**	2.08 ± 1.33
	Partially disagree	14	20.6	
	Neutral	9	13.2	
	Partially agree	6	8.8	
	Strongly agree	6	8.8	
The information about EBP during my undergraduate degree was sufficient	Strongly disagree	3	4.4	3.39 ± 1.16
	Partially disagree	14	20.6	
	Neutral	18	26.5	
	**Partially agree**	**19**	**27.9**	
	Strongly agree	14	20.6	
I know clearly how to apply these research findings to clinical practise	Strongly disagree	5	7.4	3.69 ± 1.23
	Partially disagree	7	10.3	
	Neutral	14	20.6	
	Partially agree	20	29.4	
	**Strongly agree**	**22**	**32.3**	
I understand different types of studies (study designs)	Strongly disagree	3	4.4	3.76 ± 1.19
	Partially disagree	8	11.8	
	Neutral	16	23.5	
	Partially agree	16	23.5	
	**Strongly agree**	**25**	**36.8**	
I do not understand statistical data	**Strongly disagree**	**21**	**30.9**	2.38 ± 1.22
	Partially disagree	17	25	
	Neutral	17	25	
	Partially agree	9	13.2	
	Strongly agree	4	5.9	
I am not interested in learning more about EBP	**Strongly disagree**	**39**	**57.4**	1.86 ± 1.20
	Partially disagree	10	14.7	
	Neutral	12	17.6	
	Partially agree	3	4.4	
	Strongly agree	4	5.9	

Bold values represent the largest proportion of the corresponding part. EBP, evidence-based practice; M, mean; SD, standard deviation; N, number.

**Table 3 healthcare-10-02302-t003:** Self-reported characteristics related to EBP—skills and resources.

Questions about Skills and Resources	Categories	N	%	M ± SD
I cannot conduct searches in online databases	**Strongly disagree**	**29**	**42.6**	2.08 ± 1.18
	Partially disagree	17	25	
	Neutral	11	16.2	
	Partially agree	9	13.2	
	Strongly agree	2	2.9	
I can critically assess scientific papers	Strongly disagree	6	8.8	3.39 ± 1.24
	Partially disagree	12	17.6	
	Neutral	13	17.1	
	**Partially agree**	**23**	**33.8**	
	Strongly agree	14	20.6	
I often access online database	Strongly disagree	4	5.9	3.63 ± 1.19
	Partially disagree	8	11.8	
	Neutral	17	25	
	Partially agree	19	27.9	
	**Strongly agree**	**20**	**29.4**	
I am not motivated to use EBP in my daily practise	**Strongly disagree**	**32**	**47.1**	2.20 ± 1.34
	Partially disagree	9	13.2	
	Neutral	12	17.6	
	Partially agree	11	16.2	
	Strongly agree	4	5.9	
I have computer resources and Internet access at my workplace that facilitate the use of EBP	Strongly disagree	12	17.6	3.44 ± 1.50
	Partially disagree	8	11.8	
	Neutral	9	13.2	
	Partially agree	16	23.5	
	**Strongly agree**	**23**	**33.8**	
I do not have discussions about EBP in my workplace	Strongly disagree	12	17.6	3.11 ± 1.45
	Partially disagree	15	22.1	
	Neutral	10	14.7	
	Partially agree	15	22.1	
	**Strongly agree**	**16**	**23.5**	
I ask my patients about their preferences, and I consider them in my decision-making	Strongly disagree	6	8.8	3.80 ± 1.24
	Partially disagree	4	5.9	
	Neutral	12	17.6	
	Partially agree	21	30.9	
	**Strongly agree**	**25**	**36.8**	
I inform my patients of their treatment options and consider their choices in the decision-making process	Strongly disagree	5	7.4	3.97 ± 1.13
	Partially disagree	1	1.5	
	Neutral	11	16.2	
	Partially agree	25	36.8	
	**Strongly agree**	**26**	**38.2**	
I do not use the best scientific evidence in my clinical practise	**Strongly disagree**	**29**	**42.6**	2.14 ± 1.23
	Partially disagree	15	22.1	
	Neutral	12	17.6	
	Partially agree	9	13.2	
	Strongly agree	3	4.4	

Bold values represent the largest proportion of the corresponding part. EBP, evidence-based practice; M, mean; SD, standard deviation; N, number.

**Table 4 healthcare-10-02302-t004:** Self-reported characteristics related to EBP—opinion.

Question about Opinion	Categories	N	%	M ± SD
EBP is important to my clinical practise	Strongly disagree	3	4.4	4.23 ± 1.13
	Partially disagree	4	5.9	
	Neutral	7	10.3	
	Partially agree	14	20.6	
	**Strongly agree**	**40**	**58.8**	
I do not believe that EBP improves patient care in physiotherapy	**Strongly disagree**	**47**	**69.1**	1.55 ± 0.95
	Partially disagree	8	11.8	
	Neutral	10	14.7	
	Partially agree	2	2.9	
	Strongly agree	1	1.5	
Much of my decision-making regarding the treatment of my patients incorporate EBP	Strongly disagree	4	5.9	3.51 ± 1.17
	Partially disagree	10	14.7	
	Neutral	17	25	
	**Partially agree**	**21**	**30.9**	
	Strongly agree	16	23.5	
The use of the best scientific evidence does not improve the quality of health services	**Strongly disagree**	**29**	**42.6**	2.02 ± 1.15
	Partially disagree	20	29.4	
	Neutral	10	14.7	
	Partially agree	4	8.8	
	Strongly agree	5	4.4	
An expert’s opinion in my field is the most important factor in my decision-making process	Strongly disagree	7	10.3	3.05 ± 1.07
	Partially disagree	11	16.2	
	**Neutral**	**26**	**38.2**	
	Partially agree	19	27.9	
	Strongly agree	5	7.4	

Bold values represent the largest proportion of the corresponding part. EBP, evidence-based practice; M, mean; SD, standard deviation; N, number.

**Table 5 healthcare-10-02302-t005:** Self-reported characteristics related to EBP—Barriers.

Questions about the Barriers	Categories	N	%	M ± SD
Language of scientific papers	**Strongly disagree**	**19**	**27.9**	2.66 ± 1.31
	Partially disagree	11	16.2	
	Neutral	17	25	
	Partially agree	16	23.5	
	Strongly agree	5	7.4	
Lack of quality of scientific papers	Strongly disagree	10	14.7	2.86 ± 1.11
	Partially disagree	15	22.1	
	Neutral	19	27.9	
	**Partially agree**	**22**	**32.4**	
	Strongly agree	2	2.9	
Difficulty in obtaining full-text papers	Strongly disagree	9	13.2	3.64 ± 1.34
	Partially disagree	4	5.9	
	Neutral	11	16.2	
	**Partially agree**	**22**	**32.4**	
	**Strongly agree**	**22**	**32.4**	
Lack of time	Strongly disagree	5	7.4	3.50 ± 1.13
	Partially disagree	8	11.8	
	Neutral	15	22.1	
	**Partially agree**	**28**	**41.2**	
	Strongly agree	12	17.6	
Difficulty in understanding statistics	Strongly disagree	10	14.7	3.04 ± 1.33
	**Partially disagree**	**16**	**23.5**	
	Neutral	15	22.1	
	Partially agree	15	22.1	
	Strongly agree	12	17.6	
Difficulty in understanding the results of the study	Strongly disagree	18	26.5	2.38 ± 1.15
	**Partially disagree**	**22**	**32.4**	
	Neutral	15	22.1	
	Partially agree	10	14.7	
	Strongly agree	3	4.4	
Difficulty in explaining the evidence to the patient	Strongly disagree	13	19.1	2.63 ± 1.18
	**Partially disagree**	**21**	**30.9**	
	Neutral	16	23.5	
	Partially agree	14	20.6	
	Strongly agree	4	5.9	
Applicability of research findings in clinical practise	Strongly disagree	5	7.4	3.23 ± 1.14
	Partially disagree	14	20.6	
	Neutral	18	26.5	
	**Partially agree**	**22**	**32.4**	
	Strongly agree	9	13.2	
Lack of EBP training	Strongly disagree	3	4.4	3.41 ± 1.12
	Partially disagree	13	19.1	
	Neutral	17	25	
	**Partially agree**	**23**	**33.8**	
	Strongly agree	12	17.6	
Lack of knowledge about the basics of the research	Strongly disagree	10	14.7	2.91 ± 1.19
	Partially disagree	15	22.1	
	**Neutral**	**20**	**29.4**	
	Partially agree	17	25	
	Strongly agree	6	8.8	
Lack of skills for critical appraisal	Strongly disagree	9	13.2	3.13 ± 1.22
	Partially disagree	10	14.7	
	**Neutral**	**21**	**30.9**	
	Partially agree	19	27.9	
	Strongly agree	9	13.2	
Lack of motivation for research	Strongly disagree	10	14.7	3.27 ± 1.30
	Partially disagree	9	13.2	
	Neutral	12	17.6	
	**Partially agree**	**26**	**38.2**	
	Strongly agree	11	16.2	
EBP disregards the patients’ preferences	Strongly disagree	17	25	2.33 ± 0.95
	Partially disagree	17	25	
	**Neutral**	**28**	**41.2**	
	Partially agree	6	8.8	
	Strongly agree	0	0	
Using EBP may represent a higher cost	Strongly disagree	9	13.2	2.70 ± 1.06
	**Partially disagree**	**22**	**32.4**	
	Neutral	19	17.9	
	Partially agree	16	23.5	
	Strongly agree	2	2.9	
The unfamiliarity of using online databases	Strongly disagree	15	22.1	2.57 ± 1.26
	**Partially disagree**	**22**	**32.4**	
	Neutral	15	22.1	
	Partially agree	9	13.2	
	Strongly agree	7	10.3	
Workload pressures	Strongly disagree	3	4.4	3.70 ± 1.13
	Partially disagree	9	13.2	
	Neutral	11	16.2	
	**Partially agree**	**27**	**39.7**	
	Strongly agree	18	26.5	
Lack of management support (time, encouragement, department lecture timetable, journal clubs, support continues education)	Strongly disagree	4	5.9	3.54 ± 1.23
	Partially disagree	12	17.6	
	Neutral	14	20.6	
	**Partially agree**	**19**	**27.9**	
	**Strongly agree**	**19**	**27.9**	

Bold values represent the largest proportion of the corresponding part. EBP, evidence-based practice; M, mean; SD, standard deviation; N, number.

**Table 6 healthcare-10-02302-t006:** Description of the generated scores.

Scores	Mean	Median	SD	Minimum	Maximum
**Knowledge**	34.60	35	6.27	23	45
**Skills and resources**	32.69	33	6.74	16	45
**Opinion**	19.22	20	3.47	9	25
**Barrier**	51.57	52	13.22	17	81
**Total**	138.09	139	19.79	80	184

SD, standard deviation.

**Table 7 healthcare-10-02302-t007:** Pearson’s correlation matrix between each score.

Variables	Total Score	Knowledge	Skills and Resources	Opinion	Barrier
**Total score**	1				
**Knowledge**	0.65 **0.18**	1			
**Skills and resources**	0.62 **0.24**	0.66 **0.02 ***	1		
**Opinion**	0.75 **0.05 ***	0.68 **0.02 ***	0.65 **0.04 ***	1	
**Barrier**	0.68 **0.24**	−0.02 **0.61**	−0.06 **0.51**	0.21 **0.94**	1

Bold values represent the *p*-value and * represents significance (*p* < 0.05).

**Table 8 healthcare-10-02302-t008:** Multiple regression models with (A) knowledge, (B) skills and resources, (C) opinion, and (D) barrier scores as dependent variables.

Variable	Parameter Estimates (SD)	t	*p*-Value
**A: Dependent variable: Knowledge score**			
Age	−1.31 (1.47)	−0.89	0.37
Years of experience	1.15 (0.77)	1.50	0.13
Level of education	3.04 (1.26)	2.41	0.01 *
Database usage	1.47 (0.59)	2.46	0.01 *
Model statistics	R^2^ = 0.21; F (4, 63) = 5.60, *p* < 0.001, ƒ^2^ = 0.26		
**B: Dependent variable: Skills and resources score**			
Age	1.55 (1.51)	1.02	0.30
Years of experience	0.25 (0.79)	0.32	0.74
Level of education	1.27 (1.29)	0.98	0.32
Database usage	2.34 (0.61)	3.81	<0.001 *
Model statistics	R^2^ = 0.28; F (4, 63) = 7.73, *p* < 0.001, ƒ^2^ = 0.38		
**C: Dependent variable: opinion score**			
Age	0.10 (0.82)	0.12	0.90
Years of experience	0.87 (0.43)	2.02	0.04 *
Level of education	0.26 (0.70)	0.38	0.70
Database usage	0.81 (0.33)	2.43	0.01 *
Model statistics	R^2^ = 0.19; F (4, 63) = 5.05, *p* < 0.001, ƒ^2^ = 0.23		
**D: Dependent variable: Barrier score**			
Age	2.51 (3.40)	0.73	0.46
Years of experience	1.56 (1.77)	0.88	0.38
Level of education	−5.53 (2.90)	−1.90	0.06
Database usage	−1.71 (1.38)	−1.24	0.21
Model statistics	R^2^ = 0.06; F (4, 63) = 2.16, *p* = 0.08, ƒ^2^ = 0.06		

* *p* < 0.01; SD: standard deviation.

## Data Availability

Not applicable.

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
