# Peer review of "Evidence-Based Physiotherapy Practice in Paediatric Subdiscipline: A Cross-Sectional Study in Saudi Arabia"

_healthcare, 2022, doi:10.3390/healthcare10112302_

Round 1

Reviewer 1 Report

Dear Author

This study is an important study that identifies issues in the practice of EBP in paediatrics in Saudi Arabia.

In order to improve the quality of the manuscript. After reading in depth the manuscript, I would like to make some comments and ask the authors several questions about.

Major Comments

Intoduction

1.         The majority of the description relates to EBPs in general. However, the aim of this study is 'EBPs in paediatrics'. Therefore, the significance of conducting this study should be added further.

2.         The size of the population in this study is unknown, the full number of physiotherapists in Saudi Arabia needs to be stated in the Introduction or Discussion.

3.         There is a dissociation between the abstract (L26) and the conclusions of the manuscriptL256. Either of them needs to be corrected as the one stated in the conclusion is more suitable for the text.

 Mainor Comments

1.         Sub-items in Table 1 should be left-aligned

2.         Table 3: Results should be presented in the text according to the order in the table.

3.         It is unclear how 'I understand different types of studies (study designs)' relates to EBPs. The intention of visiting this question should be stated in the Introduction or Discussion. The same applies to 'I do not understand statistical data'.

4.         Table 5: Results should be presented in the text according to the order in the table.

5.         Despite the same 'I do not understand statistical data' fingerprints in Tables 3 and 6, there are significant differences in the results. This could be related to the reliability of the questionnaire.

Author Response

Your comments were valuable and highlighted important points. Hope you will find our response to them sufficient. Please see the attachment. 

Reviewer 2 Report

I believe this study will be very beneficial in promoting quality physical therapy in Saudi Arabia.

Although this study is a survey on Saudi Arabia, I think it is very good to takes into consideration the evidence in other foreign countries.

I think that investigating the situation in Saudi Arabia will provide very useful information from an international perspective.

We look forward to further development of the survey in the future.

Please refer to below comments.

--

Please clarify why you focused on pediatric PT.

Line 43: EPB → EBP

Line 52-61: There is discrepancy in content between Line 52-56 and Line 56-61. I recommend separating paragraphs or reconsidering the sentence structure.

Line 52-61: It is recommended that the importance of EBP in the field of physical therapy be discussed.

Line 63: EPB → EBP

Line 125: Was the sample size calculated by using power analysis. Depending on the population of physical therapists, we feel that the number of PT surveyed in this study is very small.

Line 125- : How many PT is there in Saudi Arabia? In addition, it would be good to add the characteristics of physical therapists in Saudi Arabia (population, gender ratio, age, etc.) in the background.

Table1: Why do you not show male number? 

Line 134-140:There is duplication of texts and table2content. I recommend that table2 be removed and that it be specifically stated in writing.

Table 3-6: Have any statistical analyses been considered for these results? Chi-square test, etc.

Throughout the method, the amount of tables/data is very large; consider summarizing and concisely tabulating as much as possible. (e.g., explaining within the text).

I feel that you are presenting so many findings but not mentioning all of them in discussion. I also think that the considerations need to be further deepened. If possible, I recommend careful interpretation and discussion of each result.

Author Response

(The authors gave the same response as above.)

Round 2

Reviewer 1 Report

Dear Author

No further comment from me.
Thank you for correcting the manuscript.

Author Response

Thank you 

Reviewer 2 Report

Thank you very much for your careful correction.

Your correction reflects all my comments.

Author Response

Thank you.